# Pooling sputum for Xpert MTB/RIF and Xpert Ultra testing during the Covid-19 pandemic in Lao People's Democratic Republic

**Vibol Iem**[1,2], **Phonenaly Chittamany**[1], **Sakhone Suthepmany**[1], **Souvimone Siphanthong**[1], **Silaphet Somphavong**[3], **Konstantina Kontogianni**[2], **James Dodd**[2], **Jahangir Akber Mahmud Khan**[4], **Jose Dominguez**[5], **Tom Wingfield**[2,6], **Jacob Creswell**[7‡], **Luis Eduardo Cuevas**[2‡]*

**1** National Tuberculosis Control Center, Vientiane, Lao People's Democratic Republic, **2** Liverpool School of Tropical Medicine, Liverpool, United Kingdom, **3** Center of Infectiology Lao Christophe Mérieux, Vientiane, Lao People's Democratic Republic, **4** University of Gothenburg, Health Economics and Policy Unit, School of Public Health and Community Medicine, Gothenburg, Sweden, **5** Institut d'Investigació Germans Trias i Pujol, CIBER Enfermedades Respiratorias, and Universitat Autònoma de Barcelona, Barcelona, Spain, **6** WHO Collaborating Centre for Tuberculosis and Social Medicine, Department of Global Public Health, Karolinska Institutet, Solna, Sweden, **7** Stop TB Partnership, Innovations and Grants, Geneva, Switzerland

‡ JC and LEC are joint senior authors on this work.
* Luis.Cuevas@lstmed.ac.uk

**Data Availability Statement:** All relevant data are within the paper and its Supporting Information files.

## Abstract

The global Covid-19 pandemic has limited access to molecular TB diagnostics and National Programmes are struggling to maintain essential services. The pooling method (testing several samples together) could reduce the number of cartridges and staff time needed for TB diagnosis but has not been tested within the pandemic. We conducted two independent cross-sectional surveys. Pools composed of four sputum samples were tested using either Xpert-MTB/RIF or Xpert-Ultra. Pooled and individual results were compared to determine the level of agreement. Each survey included 840 participants and 210 pools. In the Xpert MTB/RIF survey, 77/81 (sensitivity 95.1%, 95%CI 87.8%-98.6%) pools containing ≥1 positive sample tested MTB-positive and 4/81 (4.9%, 95%CI 1.4%-12.2%) tested MTB-negative. All 129/129 pools containing MTB-negative samples tested MTB-negative (specificity 100%, 95%CI 97.2%-100%), with 98.1% agreement (Kappa: 0.959). In the Xpert-Ultra survey, 70/70 (sensitivity 100%, 95%CI 94.9%-100%) pools containing ≥ 1 MTB-positive sample tested MTB-positive and 140/140 (specificity 100%, 95%CI 97.4%-100%) pools containing only MTB-negative samples tested MTB-negative, with 100% agreement (Kappa: 1). Pooled testing with Xpert-MTB/RIF and Xpert-Ultra saved 38.3% and 41.7% (322/840 and 350/840, respectively) in cartridge costs alone. The pooling method with Xpert-MTB/RIF and Xpert-Ultra has similar performance to individual testing and can reduce the number of cartridges needed. These efficiencies can facilitate maintenance of stocks and sustain essential services as countries face difficulties for laboratory procurement during the pandemic and will provide cost and time savings post-pandemic.

**Funding:** TB REACH, Stop TB Partnership grant supported by Global Affairs Canada (STBP/TBREACH/GSA/2020-04, awarded to LEC); the National Institute for Health Research Health Protection Research Unit in Emerging and Zoonotic Infections, the Centre of Excellence in Infectious Diseases Research, and the Alder Hey Charity (awarded to LEC); the Global Fund to Fight AIDS, Tuberculosis and Malaria (LAO-T-GFMOH awarded to Lao's PDR Ministry of Health grant). Costs for consumables and laboratory staff were covered by the National Tuberculosis Control Center though its operational budget from the Ministry of Health of Lao PDR (awarded to the NTC). TW is supported by grants from: the Wellcome Trust, UK (209075/Z/17/Z); the Medical Research Council, Department for International Development, and Wellcome Trust, UK (Joint Global Health Trials, MR/V004832/1), the Medical Research Council, UK (MR/V028618/1); the Academy of Medical Sciences, UK; and the Swedish Health Research Council, Sweden. The funders had no role in study design, data collection and analysis, nor decision to publish, or preparation of the manuscript.

**Competing interests:** The authors have no conflicts of interest to declare.

## Introduction

Tuberculosis (TB) is a major cause of morbidity and death worldwide, with an estimated 10 million people falling ill and 1.4 million deaths occurring in 2019 alone [1]. Despite its public health importance, three million people with TB are missed by national TB programmes (NTPs) every year [1], due to accessibility barriers, and diagnosis, treatment and notification gaps [2].

The World Health Organization (WHO) recommends testing individuals with presumptive TB with molecular assays as the first test for bacteriological confirmation [3]. These tests include Xpert MTB/RIF and Xpert MTB/RIF Ultra (Xpert Ultra), which are automated and simultaneously detect *Mycobacterium tuberculosis* complex and markers of rifampicin resistance [4] and the latter, is currently recommended in preference to Xpert MTB/RIF, based on its increased sensitivity, which improves the detection of paucibacillary forms of TB [5]. However, despite major international initiatives to increase the use of molecular assays, the majority of TB diagnoses in low and middle income countries are based on smear microscopy, which has lower sensitivity [6] and does not detect drug resistance, but is locally available and has lower costs than molecular assays [7].

Lao's People's Democratic Republic (PDR) had an estimated incidence of 155 people with TB per 100,000 population in 2019 and has improved TB case detection in recent years, with the number of people detected increasing from 44 per 100,000 population in 2000 to 95 per 100,000 in 2019. Lao PDR has a low prevalence of rifampicin resistance-TB with 1.2% (95% CI: 0.5–2.0%) and 4.1% (95% CI: 0–9.6%) among new cases and previously treated cases, respectively, and a low prevalence of multi-drug resistance TB (MDR-TB) with 0.5% (95% CI: 0–1.0%) and 2.3% (95% CI: 0–6.7%), respectively [8]. Improved detection is partly due to concentrated efforts to identify people with presumptive TB and an increased use of Xpert as the first test for diagnosis, with 66% of people with presumptive TB tested with Xpert MTB/RIF in 2019 [9], and the country aims to provide universal Xpert testing from 2021 onwards. These ambitious targets, however, would require considerable increases in cartridges, GeneXpert instruments and human resources, resulting in higher costs.

At the end of 2019, coronavirus disease-19 (Covid-19) caused by the Severe Acute Respiratory Syndrome Coronavirus 2 (SARS-CoV-2) turned into an epidemic that triggered chaos in hospitals and primary care services [10]. On the 30th January 2020 the WHO declared this outbreak a Public Health Emergency of International Concern [11] and countries with limited laboratory resources, such as Lao PDR, were requested to share existing GeneXpert platforms for both COVID-19 and TB testing [12]. Lockdowns and reassignments of health personnel and equipment away from TB created a devastating impact on the performance of NTPs, especially in low and middle income countries [13].

To address these challenges, we evaluated whether combining specimens of four individuals with presumptive TB in a pool and testing the pool using either Xpert MTB/RIF or Xpert Ultra would result in the same accuracy as testing samples individually, and estimated whether the approach would result in assay cost savings. In the pooling method, when a pool tests positive, all individual samples included in that pool are re-tested individually to identify the positive sample(s), while if the pool tests negative, it is assumed all samples included are negative. Pooling could then be used to test larger numbers of people with the same number of cartridges, thus increasing the efficiency of Xpert-based testing in locations with limited resources [14].

## Materials and methods

The study took place at Mahosot hospital and nine district health facilities in Vientiane, the capital of Lao PDR, and included individuals presenting to the TB diagnostic services with a

diagnosis of presumptive TB. The national technical guidelines (NTG) define a person with presumptive TB as an individual with cough for 2 weeks duration or with two or more symptoms including cough, hemoptysis, weight loss, fever, night sweats, tiredness, chest pain, dyspnea, or the presence of chest X-ray abnormalities suggestive of TB. The guidelines recommend requesting one sputum sample from all individuals with presumptive TB and to examine the sample using Xpert MTB/RIF or Xpert Ultra, as available. Sputum samples were collected on the spot at the time individuals presented to the clinics and were sent to the National TB Reference Laboratory at the National TB Control Center in Vientiane. The study consisted of two separate cross-sectional surveys, with the first taking place from March to May 2020 (called the 2020 survey) and the second survey from January to March 2021 (the 2021 survey). All individual and pooled samples in the 2020 survey were tested using Xpert MTB/RIF and all individual and pooled samples in the 2021 survey were tested using Xpert Ultra, at a time when Xpert Ultra had become available in the country.

All Xpert MTB/RIF and Xpert Ultra testing followed the manufacturer's instructions. Briefly, the sample reagent was added to the sample at a 2:1 ratio, mixed on a vortex and left at room temperature for 10 minutes. The sample was then vortexed again and left to stand for a further five minutes. Two ml of the sputum sample were then loaded into the Xpert cartridge for individual testing [15] and the remnant of the specimens were grouped into batches of four to prepare the pools for testing. Pools of four were selected because in settings such as Lao PDR the proportion of positive samples is between 1% and 30% and this pool size may be close to optimal [16]. The pools of specimens were created using consecutive samples without knowing the results of the individual tests. An equal volume of 0.5 ml of each of the four individual pre-treated samples was added to a new cup and the cup was vortexed and loaded into a new Xpert cartridge (Fig 1). Pooled samples were tested in batches, independently of individual tests.

Xpert trace calls were considered as MTB-positive as per the NTG, and patients were re-tested with a new sample to determine the rifampicin resistance status. Individual Xpert results were communicated back to the diagnostic centers and were the only Xpert test result used for patient management, while pooled sputum results were only used for research purposes and were not reported to the clinicians nor the patients.

## Statistical analysis

Categorical data were summarized using descriptive statistics, with chi-squared tests used to test for statistically significant differences, where appropriate. Results obtained with the pooled samples were compared with the four individual results for both Xpert MTB/RIF and Xpert Ultra. The agreement of the pooled and individual tests was tested using kappa statistics. We compared the CT values and the Xpert grades (*trace*, *very low*, *low*, *medium*, and *high*) for individual tests for concordance with the results from pools containing a single positive test. Cost differences were calculated on the basis of the number of cartridges that would have been required to test all specimens when using either a pooled or an individual testing strategy. We then modeled the potential cost savings from our results for testing 1,000 consecutive individuals with Xpert at the FIND negotiated cartridge cost of USD 9.98 [17] and calculated the additional people tested for TB when using 1,000 cartridges with the pooling method.

The datasets used and/or analysed during the current study are available from the corresponding author on reasonable requests for guideline development and systematic reviews.

Need for ethical approval and informed consent were waived by the National TB Control Center of Lao PDR and the Liverpool School of Tropical Medicine Research Ethics Committee, UK (Ethical waiver 20–037).

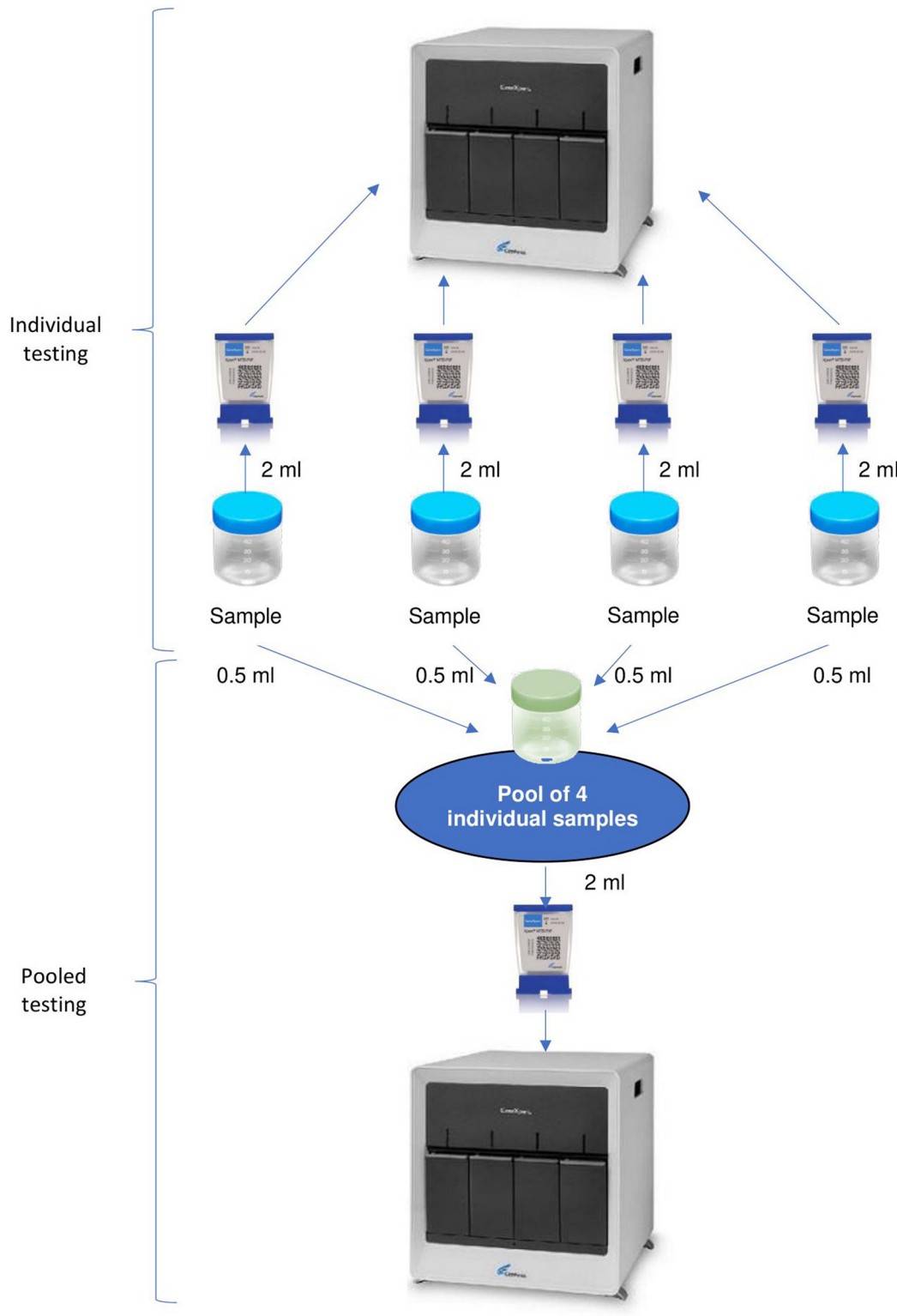

**Fig 1. Flow diagram of the sputum processing.**

## Results

A total of 1680 participants were included, with 840 participants tested with Xpert MTB/RIF in the 2020 survey and 840 with Xpert Ultra in the 2021 survey (S1 Data).

### Xpert MTB/RIF survey

In the 2020 survey, 491/840 (58.5%) participants were male and 349 (41.5%) female and 102/840 (12.1%) were Xpert MTB/RIF MTB-positive (Table 1). Males were more likely to be Xpert

**Table 1. Baseline characteristics of participants with single and pooled Xpert results.**

| | | Xpert | |
| --- | --- | --- | --- |
| | | MTB/RIF | Ultra |
| | | All | All |
| | | n = 840 (%) | n = 840 (%) |
| **Sex** | Male | 491 (58.5) | 500 (59.5) |
| | Female | 349 (41.5) | 340 (40.5) |
| **Age (Mean (sd, range))** | | 49 (19.3, 1–98) | 51 (19.6, 1–96) |
| **Age group (years)** | <35 | 237 (28.2) | 210 (25.0) |
| | 35–54 | 238 (28.3) | 207 (24.6) |
| | > = 55 | 365 (43.5) | 423 (50.4) |
| **Individual Xpert Result** | Detected | 102 (12.1) | 100 (11.9) |
| | Not detected | 738 (87.9) | 740 (88.1) |
| | Male MTB detected | 72/491 (14.7) | 59/500 (11.8) |
| | Female MTB detected | 30/349 (8.6) | 41/340 (12.1) |
| | | **n = 102** | **n = 100** |
| **MTB Grade** | Trace | NA | 14 (14.0) |
| | Very low | 17 (16.7) | 10 (10.0) |
| | Low | 22 (21.6) | 27 (27.0) |
| | Medium | 44 (43.1) | 9 (9.0) |
| | High | 19 (18.6) | 40 (40.0) |
| | | **n = 102** | **n = 100** |
| **Rif Resistance** | Detected | 0 (0.0) | 1 (1.0) |
| | Not detected | 102 (100.0) | 85 (85.0) |
| | Indeterminate | 0 (0.0) | 14 (14.0) |
| | | **n = 210** | **n = 210** |
| **Samples tested in the pool** | ≥ 1 MTB-positive sample | 81 (38.6) | 70 (33.3) |
| | 4 MTB-negative samples | 129 (61.4) | 140 (66.7) |
| **Pooled Xpert MTB result** | Detected | 77 (36.7) | 70 (33.3) |
| | Not detected | 133 (63.3) | 140 (66.7) |
| | | **n = 77** | **n = 70** |
| **Pooled MTB Grade** | Trace | NA | 11 (15.7) |
| | Very low | 19 (24.7) | 12 (17.1) |
| | Low | 27 (35.1) | 24 (34.3) |
| | Medium | 24 (31.2) | 6 (8.6) |
| | High | 7 (9.1) | 17 (24.3) |
| | | **n = 77** | **n = 70** |
| **Pooled Rif Resistance** | Detected | 0 (0.0) | 2 (2.9) |
| | Not detected | 77 (100.0) | 57 (81.4) |
| | Indeterminate | 0 (0.0) | 11 (15.7) |

**Table 2. Distribution of positive individual samples among pooled results, by Xpert test.**

| | Number of positive Xpert results included in a pool | | | | | |
| --- | --- | --- | --- | --- | --- | --- |
| | All negative n (%) | One positive n (%) | Two positive n (%) | Three positive n (%) | Four positive n (%) | All n (%) |
| **Pooled Xpert MTB/RIF** | **129** | **62** | **17** | **2** | **0** | **210** |
| Detected | 0 (0%) | 58 (93.5%) | 17 (100%) | 2 (100%) | 0 (0%) | 77 (36.7%) |
| Not detected | 129 (100%) | 4 (6.5%) | 0 (0%) | 0 (0%) | 0 (0%) | 133 (63.3%) |
| **Pooled Xpert Ultra** | **140** | **45** | **20** | **5** | **0** | **210** |
| Detected | 0 (0%) | 45 (100%) | 20 (100%) | 5 (100%) | 0 (0%) | 70 (33.3%) |
| Not detected | 140 (100%) | 0 (0%) | 0 (0%) | 0 (0%) | 0 (0%) | 140 (66.7%) |

MTB-positive than females (72/491 (14.7%) and 30/349 (8.6%), respectively, but these differences were not statistically significant, p-value > 0.1).

Individual samples were tested in 210 pools. Of these, 81 (38.6%) pools contained at least one Xpert MTB-positive sample and 129 (61.4%) had only Xpert MTB-negative samples. Sixty-two (75%) of the 81 pools with MTB-positive samples contained only one MTB-positive sample, 17 (21%) contained two MTB-positive and two (2.5%) pools contained three MTB-positive samples (Table 2). Seventy-seven (sensitivity 95.1%, 95%CI 87.8% - 98.6%) of the 81 pools with MTB-positive samples tested Xpert MTB-positive in the pooled assay and four (4.9%, 95%CI 1.4% - 12.2%) tested negative. None (0%) of the 129 pools containing only MTB-negative samples returned a pooled Xpert MTB-positive result, resulting in 100% (95% CI 97.2% - 100%) specificity. The agreement between the individual and pooled approaches was 98.1% (Kappa: 0.959).

Thirteen individual samples tested with Xpert MTB/RIF had *very low*, 11 *low*, 25 *medium* and 13 *high* MTB-grades. The grades for the pools containing single MTB-positive samples are shown in Fig 2A. The MTB-grade was the same in 29/62 (46.8%) individual and pooled tests and discrepant in 33/62 (53.2%). The discrepancies were always in the same direction, with the pooled MTB-grade being one grade lower than the individual MTB-grade in 28/33 (85%) and two steps lower in five (15%) of the discrepant samples. The four pools testing MTB-negative by the pooled Xpert MTB/RIF but positive by the individual test had *very low* individual MTB-grades (Fig 2A).

The median CT values of the Xpert MTB/RIF probes are shown in Table 3. Individual A-E probes had median CTs ranging from 20.4 to 21.9. Pooled assays had higher CT values with CT values ranging from 23.8 to 25.0, with difference between individual and pooled assays ranging from 2.8 to 3.6 CTs. Lastly, none of the 840 samples tested were Xpert RIF-positive or indeterminate. Consequently, all pools with MTB-positive samples contained only RIF-negative samples and none of them reported pooled RIF-positive results.

## Xpert Ultra survey

In the 2021 survey, 500/840 (59.5%) participants were male and 340/840 (40.5%) female and 100/840 (11.9%) were Xpert Ultra MTB-positive (Table 1). Males and females were equally likely to be Xpert Ultra MTB-positive (59/500 (11.9%) and 41/340 (12.1%), respectively, p-value > 0.1, Table 1). Individual samples were tested in 210 pools and of these, 70 contained at least one MTB-positive sample and 140 contained only MTB-negative samples. Among the 70 pools with MTB-positive samples, 45 contained one MTB-positive, 20 contained two MTB-positive and five contained three MTB-positive samples, as shown in Table 2. All 70/70 pools containing at least one MTB-positive sample tested Xpert MTB-positive in the pooled assay, resulting in 100% (95%CI 94.9% - 100%) sensitivity, and all 140/140 pools containing only MTB-negative samples tested Xpert MTB-negative (specificity 100%, 95%CI 97.4% - 100%)

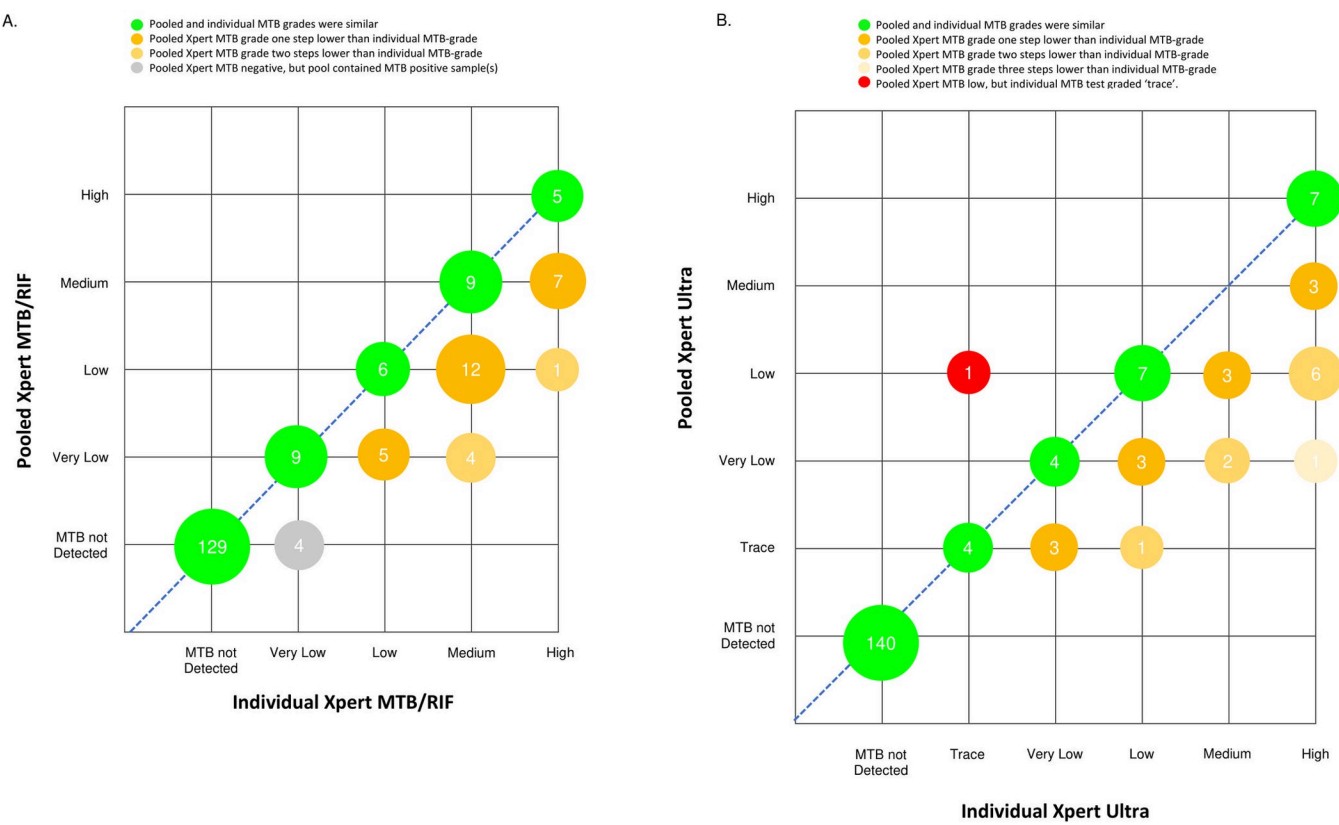

**Fig 2.** Correlation of individual and pooled (A) Xpert MTB/RIF and (B) Ultra grades (positive pools only include those with one individual Xpert MTB-positive sample).

and 100% agreement (Kappa: 1). Seven individual samples tested with Xpert Ultra had *very low*, 11 *low*, 5 *medium* and 17 *high* MTB-grades, with 5 samples reporting *trace* results, as shown in Fig 2B. The MTB-grades coincided in 22/45 (48.9%) individual and pooled tests and was discrepant in 23/45 (51.1%). Similar to the Xpert MTB/RIF survey, in all but one of the discrepancies the pooled MTB-grade was lower than the MTB-grade of the individual test (Fig 2B). The CT values of the Xpert Ultra probes are shown in Table 3. Individual probes (IS1081/ IS6110 and rpoB1-B4) had median CTs ranging from 16.7 to 21.4 and pooled CTs ranged from 18.1 to 24.1, with a CT difference between individual and pooled assays ranging from 1.3 to 2.7. Only one (1%) of the 100 MTB-positive samples was RIF-positive, 14/100 (14%) were RIF-indeterminate and 85/100 (85%) RIF-negative. Among the 70 pools containing MTB-positive samples, 1/70 (1.4%) had the single RIF positive sample, 12/70 (17.1%) contained a single RIF-indeterminate and 1/70 (1.4%) two RIF-indeterminate samples. The RIF-positive sample tested RIF-positive in the pooled test. Four of the 12 pools containing single RIF-indeterminate samples tested pooled RIF-indeterminate and eight tested RIF-negative. The pool containing two RIF-indeterminate samples tested pooled RIF-indeterminate. Of the 56 pools containing solely RIF-negative samples, 49/56 (88%) tested pooled RIF-negative, 6/56 (11%) pooled RIF-indeterminate and 1/56 (2%) RIF-positive.

## Xpert MTB/RIF and Xpert Ultra savings

The number of cartridges required to test the 840 individuals using the pooling method was estimated for both surveys. Testing 210 pools with Xpert MTB/RIF required 210 cartridges,

**Table 3. Median CT values of individual and pooled Xpert MTB/RIF and Xpert Ultra probe results.**

| | Xpert MTB RIF | | | | |
|---|---|---|---|---|---|
| | Individual results n = 102 | | Pooled results n = 77 | | |
| Probe | CT Median IQ range | Min-Max | CT Median IQ range | Min-Max | ΔCT |
| Probe D | 21.5 (18.1, 27.1) | 11.9, 35.4 | 24.9 (20.6, 28.7) | 13.2, 36.3 | 3.4 |
| Probe C | 20.7 (17.6, 26.3) | 11.5, 34.1 | 24.3 (19.8, 28.0) | 11.8, 33.5 | 3.6 |
| Probe E | 22.0 (18.6, 27.2) | 12.8, 36.6 | 25.0 (21.0, 29.9) | 13.6, 36.8 | 3.0 |
| Probe B | 21.7 (18.5, 26.5) | 12.6, 33.8 | 24.5 (20.9, 27.8) | 13.3, 34.3 | 2.8 |
| Probe A | 20.4 (16.9, 25.8) | 11.2, 34.0 | 23.8 (19.9, 28.5) | 11.7, 34.6 | 3.4 |
| | Xpert Ultra | | | | |
| | Individual results n = 100 | | Pooled results n = 70 | | |
| Probe | CT Median IQ range | Min-Max | CT Median IQ range | Min-Max | ΔCT |
| Probe IS1081/ IS6110 | 16.8 (16.2, 21.3) | 15.9, 31.0 | 18.1 (16.2, 22.2) | 16.0, 30.0 | 1.3 |
| Probe rpoB1 | 18.3 (17.4, 22.7) | 0.0, 36.5 | 20.7 (17.9, 25.7) | 0.0, 39.5 | 2.4 |
| Probe rpoB2 | 18.2 (17.4, 22.5) | 0.0, 36.0 | 20.6 (17.7, 25.6) | 0.0, 35.7 | 2.4 |
| Probe rpoB3 | 19.8 (18.6, 24.6) | 0.0, 37.5 | 21.9 (19.0, 27.1) | 0.0, 37.7 | 2.1 |
| Probe rpoB4 | 21.4 (20.3, 26.4) | 0.0, 37.8 | 24.1 (20.6, 28.8) | 0.0, 39.4 | 2.7 |

and 77 were MTB-positive. The MTB-positive pools required re-testing the individual samples to identify the positive sample/s in the pool, and this required 308 (77x4) additional test and a total of 518 Xpert MTB/RIF cartridges. The pool method therefore resulted in a saving of 322/ 840 (38.3%) cartridges (840–518). Similarly, the pooling method required 210 cartridges to test in pools and 280 additional cartridges to test the individual samples of the 70 positive pools (70X4), resulting in a total of 490 (210 + 280) Xpert Ultra cartridges. The pooling method therefore would result in a saving of 350 (840–490, or 41.7%, n = 350/840) cartridges. The results of the extrapolation to illustrate the cartridge savings achieved when screening 1,000 consecutive individuals and to the number of individuals that could be tested with a fixed number of 1,000 cartridges are shown in Table 4. Cartridge costs for testing 1,000 individuals would amount USD 9,980, and the pooling method would cost USD 6,158 and USD 5,818 for Xpert MTB/RIF and Xpert Ultra, respectively, resulting in USD 3,822 and USD 4,161 savings, respectively. Alternatively, given its efficiency, using the pooling method with a fixed number

**Table 4. Cost and diagnostic savings to screen 1000 consecutive patients using the pooling method and number of patients that could be tested with 1000 Xpert MTB/RIF and Xpert Ultra cartridges.**

| | | Xpert MTB/RIF | Xpert Ultra | Pooling Xpert MTB/RIF | Pooling Xpert Ultra |
|---|---|---|---|---|---|
| **Number of individuals tested** | | 1000 | 1000 | 1000 | 1000 |
| | **Sensitivity** | reference | reference | 95.1%* | 100%* |
| | **specificity** | reference | reference | 100%* | 100%* |
| | **Proportion positive** | 12.1%* | 11.1%* | 36.7% pools | 33.3% pools |
| | **Bacteriologically confirmed** | 121 | 111 | 115 | 111 |
| | **Cartridges required** | 1000 | 1000 | 617 | 583 |
| | **Cartridge costs (USD)** | 9,980 | 9,980 | 6,158 | 5,818 |
| | **Cartridge savings (USD)** | 0 | 0 | **3822 (38.3%)** | **4161 (41.7%)** |
| **Number tested with 1000 cartridges** | | | | | |
| | **Number tested** | 1000 | 1000 | 1620 | 1715 |
| | **Cartridge cost per patient (USD)** | 9.98 | 9.98 | **6.16** | **5.80** |

* Assumes pools of 1:4; proportion positive taken from the surveys' findings.

of 1,000 Xpert MTB/RIF and Xpert Ultra cartridges would allow testing 1,620 and 1,715 patients, reducing the effective cartridge cost per individual screened to USD 6.16 and USD 5.80, respectively.

## Discussion

This is the first report directly comparing testing pooling samples for TB using the Xpert MTB/RIF and Xpert Ultra in the same population setting and study methods. Samples for the surveys were collected and tested at the time the country had implemented quarantine measures to reduce the spread of SARS-CoV-2 infections and staff had been re-deployed in response to the Covid-19 pandemic and thus was conducted at a time when human resources were strained.

Our study adds to the emerging body of evidence that the pooling methods for testing with molecular assays can improve the efficiency of testing for TB, potentially enabling the screening and testing of larger numbers of people more cost-effectively. Our findings confirm that there is a good correlation between the results of the individual and pooled tests, with a low frequency of false-negative results and a high degree of specificity. Our findings support previous studies indicating that pooled Xpert MTB/RIF detects about 95% of MTB-positive samples and that pooled Xpert Ultra can yield full agreement between individual and pooled Xpert Ultra testing, as previously reported from Cambodia [18]. The higher agreement of Xpert Ultra is likely due to its higher sensitivity [19, 20], as its limit of detection (15.6 cfu/ml) [21] is lower than for Xpert MTB/RIF's (131 cfu/ml) [22], thus reducing the risk of the diluted TB DNA falling below the detection limit. All false Xpert MTB-negative results occurred among individual samples containing high CT values that were graded MTB-*very low*, which corresponded to the increasing CT values of the individual probes. Given the complete agreement between pooled and individual testing with Xpert Ultra, countries with limited testing resources could consider using the pooling method as a routine practice. Lao's NTC will phase out Xpert MTB/RIF once stocks are depleted, and will replace it with Xpert Ultra from 2022. The program is considering the adoption of pooling for TB once endorsed by WHO guidance.

It is important to note that our study was conducted among adults with a low prevalence of HIV (0.17%) [23] and that very few participants had dual TB-HIV co-infections, as only 5% of new TB cases in Lao PDR occur among HIV-infected individuals [9]. Individuals with HIV often present with paucibacillary TB and systematic reviews in HIV prevalent settings have reported that the sensitivity of pooled testing may be lower [24], with a higher sensitivity achieved when testing with Xpert Ultra (87.6%, 95%CI 75.4–94.1%) than with Xpert MTB/RIF (74.9%, 95%CI 58.7–86.2) [25]. Similarly our findings may be different to those observed in studies conducted during active TB case finding interventions, where the proportion of individuals with positive tests is much lower (typically less than 5%) and paradoxically patients may be identified at very early or late stages of the disease [26, 27] and therefore further studies are needed among populations with high HIV prevalence and in locations where the proportion of individuals testing positive is low.

In terms of specificity, our results confirm the high specificity of pooled testing for both Xpert MTB/RIF and Xpert Ultra, with none of the MTB-negative samples becoming positive in the pooled tests. Some studies however have reported a slightly lower specificity [21, 28], which may be attributed to the increased manipulation of samples resulting in an increased risk of contamination and labelling errors and these varying results may reflect the competency and dedicated time available of laboratory staff for sample processing.

Our results also confirm that testing for RIF resistance in the pooled assays is unreliable for Xpert Ultra, as four pools containing individual Xpert RIF-indeterminate samples tested RIF-

negative when tested in a pool and pools containing only RIF-negative samples tested pooled RIF-indeterminate. The same issues have been reported for Xpert MTB/RIF, but all samples tested in our study were RIF-negative. False-positive rifampicin resistance is not unusual in paucibacillary samples [29, 30], and more than half of Xpert Ultra false-positive rifampicin resistance results were obtained from individuals with MTB trace results [28]. However, since the pooling method requires repeating individually all samples from pools testing MTB-positive, this issue would not have misclassified individuals in routine practice.

The pooling method resulted in savings ranging from 38%-42% in cartridge costs, allowing testing more patients with a limited number of cartridges. Over the course of a year, potential savings from such an approach are large, even with a single machine, and many more people would be tested and diagnosed using the pooling method. In our setting, 620–715 (60–70%) additional TB patients could be tested with the same cost of resources, which would facilitate closing the country-wide testing gap. Cartridges and time savings however are directly related to the proportion of pools that are positive, and this proportion would change with TB prevalence and populations tested. Savings therefore maybe larger in locations with low prevalence and during active case finding, when the proportion of pools testing positive may be lower. The savings presented here therefore may underestimate actual savings. Moreover, we did not estimate other savings, such as staff time, electricity, overhead costs, and costs to patients' and their carers. The pooling method therefore can be particularly important at a time when procurement and importation of laboratory consumables is limited due to the pandemic, and when staff had been re-deployed to SAR-CoV-2 testing.

## Conclusions

The pooling method has high sensitivity and specificity for both Xpert MTB/RIF and Xpert Ultra, with the latter resulting in full agreement between individual and pooled testing. Pooled testing resulted in significant cartridge savings and facilitated more efficient testing within the pandemic, when financial resources are stretched, and the health system is strained. These promising results call for more studies to assess the potential of the pooling method in populations with low TB prevalence, such as outreach active case finding campaigns, where the proportion of people with bacteriologically confirmed TB is usually lower, as it could result in significantly higher savings. The pooling method would support the WHO End TB strategy, urging countries to expand access to rapid molecular tests for the detection of TB. In a context where countries may experience stock-outs or delays in laboratory commodities procurement due to the Covid-19 lockdown, pooling may be the optimal diagnostic option for individuals with presumptive TB.

## Supporting information

**S1 Data. Study dataset.**
(XLSX)

## Acknowledgments

The authors would like to thank the National Tuberculosis Control Center of Lao PDR for facilitating access to samples and cartridges to carry out the study in an accelerated timeline. We are grateful to the laboratory technicians of the National TB Reference Laboratory of Lao PDR for the careful management of samples and the additional work on top of their busy schedule, especially during the Covid-19 crisis. We also thank Jim Read and the Global Health Trial Unit, LSTM for generating databases and data curation.

## Author Contributions

**Conceptualization:** Vibol Iem, Konstantina Kontogianni, Jahangir Akber Mahmud Khan, Jacob Creswell, Luis Eduardo Cuevas.

**Data curation:** Vibol Iem, Konstantina Kontogianni, Tom Wingfield, Luis Eduardo Cuevas.

**Formal analysis:** Vibol Iem, Konstantina Kontogianni, James Dodd, Jose Dominguez, Tom Wingfield, Jacob Creswell, Luis Eduardo Cuevas.

**Funding acquisition:** Luis Eduardo Cuevas.

**Methodology:** Vibol Iem, Konstantina Kontogianni, Jahangir Akber Mahmud Khan, Jacob Creswell, Luis Eduardo Cuevas.

**Project administration:** Vibol Iem.

**Supervision:** Vibol Iem, Luis Eduardo Cuevas.

**Writing – original draft:** Vibol Iem, Jose Dominguez, Tom Wingfield, Jacob Creswell, Luis Eduardo Cuevas.

**Writing – review & editing:** Vibol Iem, Phonenaly Chittamany, Sakhone Suthepmany, Souvimone Siphanthong, Silaphet Somphavong, Konstantina Kontogianni, James Dodd, Jahangir Akber Mahmud Khan, Jose Dominguez, Tom Wingfield, Jacob Creswell, Luis Eduardo Cuevas.

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
