## [Decision Letter · Decision Letter 0]

2 Nov 2021

PGPH-D-21-00728

Pooling sputum for Xpert MTB/RIF and Xpert Ultra testing during the Covid-19 pandemic in Lao People's Democratic Republic

Dear Dr. Iem,

Thank you for submitting your manuscript to PLOS Global Public Health. After careful consideration, we feel that it has merit but does not fully meet PLOS Global Public Health’s publication criteria as it currently stands. Therefore, we invite you to submit a revised version of the manuscript that addresses the points raised during the review process.

We look forward to receiving your revised manuscript.

Kind regards,

Miguel Angel Garcia-Bereguiain, PhD

Academic Editor

Journal Requirements:

1. Thank you for stating "Informed consent and ethical waivers were obtained from the National TB Control Center of Lao PDR and the Liverpool School of Tropical Medicine Research Ethics Committee, UK (Ethical waiver 20-037)." Please clarify whether these ethics committees specifically approved your research, or whether they waived the need for ethics approval and the reason why approval was waived. 

In addition, please provide additional details regarding participant consent. In your Methods, please ensure that you have specified:

 - whether consent was obtained from participants

 - whether consent was informed 

 - what type of consent you obtained (for instance, written or verbal, and if verbal, how it was documented and witnessed). 

 - if your study included minors, state whether you obtained consent from parents or guardians. 

 - if the need for consent was waived by the ethics committee, please include this information.

2. We do not publish any copyright or trademark symbols that usually accompany proprietary names, eg (R), (C), or TM  (e.g. next to drug or reagent names). Therefore please remove all instances of trademark/copyright symbols throughout the text, including Xpert® on reference 29.

3. Please amend your detailed Financial Disclosure statement. This is published with the article, therefore should be completed in full sentences and contain the exact wording you wish to be published.

i). State the initials, alongside each funding source, of each author to receive each grant.

ii). State what role the funders took in the study. If the funders had no role in your study, please state: “The funders had no role in study design, data collection and analysis, decision to publish, or preparation of the manuscript.”

4. In the online submission form, you indicated that "The datasets used and/or analysed during the current study are available from the corresponding author on reasonable requests for guideline development and systematic reviews."

Additional Editor Comments (if provided):

Dear Dr Iem,

After reading your manuscript and the comments made by the 3 reviewers, I believe that your work is potentially suitable for publication. However, some concerns and comments have been made by some of the reviewers. Please, address those comments and improve the manuscript accordingly and submit the revise version of the manuscript and a letter addressing a point by point answers to reviewers.

Sincerely

Miguel Angel Garcia Bereguiain, PhD.

Reviewers' comments:

Reviewer's Responses to Questions

**Comments to the Author**

1. Does this manuscript meet PLOS Global Public Health’s publication criteria? Is the manuscript technically sound, and do the data support the conclusions? The manuscript must describe methodologically and ethically rigorous research with conclusions that are appropriately drawn based on the data presented.

Reviewer #1: Yes

Reviewer #2: Yes

Reviewer #3: Yes

2. Has the statistical analysis been performed appropriately and rigorously?

Reviewer #1: Yes

Reviewer #2: Yes

Reviewer #3: Yes

3. Have the authors made all data underlying the findings in their manuscript fully available (please refer to the Data Availability Statement at the start of the manuscript PDF file)?

Reviewer #1: Yes

Reviewer #2: Yes

Reviewer #3: Yes

4. Is the manuscript presented in an intelligible fashion and written in standard English?

Reviewer #1: Yes

Reviewer #2: Yes

Reviewer #3: Yes

5. Review Comments to the Author

Reviewer #1: The paper is very interesting and the researchers have clearly described their methods and statistical analysis. They have written the results in a very descriptive way and provided sound arguments on the discussion section.

Reviewer #2: Manuscript complies PLOS Global Public Health publication criteria and statistical analysis are appropiate for the type of study that was performed. Proper use of English that communicates easily the findings in the study.

Reviewer #3: This paper is well-written overall. The topic is very applicable in the current pandemic situation, and the conclusions are strongly supported by the data presented. I have only minor suggestions to improve the comprehension and impact.

1. General comment throughout the text: report the n/N, followed by the percent. Example, in line 41: 70/70 (100% sensitivity).

2. Paragraph starting on line 57: include more background on Xpert, Ultra, and RIF-resistance. Note that smear microscopy does not include detection of drug resistance.

3. Paragraph starting on line 67: include the prevalence of drug resistance in Lao.

4. Explain earlier in the text how if any pool tested positive, the four individual samples would need to be re-tested. This is only mentioned in the Results (line 213).

5. Add more detail to Figure 1, such as the sample volumes going to pooled vs individual testing.

6. In Table 1, there are differences in demographic and Xpert results between the two groups (2020 and 2021 surveys). Have you tested whether these are significant? Was there any difference in the populations tested between years? Xpert Ultra is more sensitive, which would explain some of the differences. But would not explain why the proportion of male/female MTB detected would switch?

7. Please report n/N (%) consistently throughout the Results section.

8. Add a legend for Figure 2, including labelling the colors used for different groups.

9. The findings for how many additional patients could be tested with the pooling method are very strong, and suggest emphasizing this point more. Could also state it as 620-715 (60-70%) additional TB patients could be tested with the same cost of resources. And could even take it one step further and apply that to closing the country-wide testing gap if the data is available.

10. Now that Xpert Ultra is available, will the standard Xpert MTB/RIF cartridges still be used in Lao? May be good to mention in the Discussion.

6. PLOS authors have the option to publish the peer review history of their article (what does this mean?). If published, this will include your full peer review and any attached files.

**Do you want your identity to be public for this peer review?** For information about this choice, including consent withdrawal, please see our Privacy Policy.

Reviewer #1: No

Reviewer #2: No

Reviewer #3: No

---

## [Editor Report · Decision Letter 1]

10 Dec 2021

PGPH-D-21-00728R1

Pooling sputum for Xpert MTB/RIF and Xpert Ultra testing during the Covid-19 pandemic in Lao People's Democratic Republic

Dear Dr Iem,

The manuscript has been substantially improved following reviewers and editorial comments. However, I realized than some minor comments made by reviewer 1 were not addressed on your response letter.

In my response letter, those comments did not show up as the were uploaded as a word file. Alhtough I believe you could access this file in the submission system, please find below those comments.

There are very minor comments so simply addressed them prior to final acceptance of your manuscript.

And sorry for this inconvenience.

We look forward to receiving your revised manuscript.

Kind regards,

Miguel Angel Garcia-Bereguiain, PhD

Academic Editor

Journal Requirements:

Additional Editor Comments (if provided):

Dear Dr Iem,

The manuscript has been substantially improved following reviewers and editorial comments. However, I realized than some minor comments made by reviewer 1 were not addressed on your response letter. On my response letter, those comments did not show up as the were uploaded as a word file. Alhtough I believe you could access this file in the submission system, please find below those comments. There are very minor comments so simply addressed them prior to final acceptance of your manuscript. And sorry for this inconvinience.

Reviewer’s comments

Line 92- please indicate the individuals as adults (as the study was done in adults)

Line 139-why did you need ethical waiver because the usual way is to ask for informed consent for each individual during enrolment to the study?

Line 145-46-the sentence mentioning that males are likely to be xpert positive than females, does it have any explanation? and p<0.1 . I don’t think it is worth to mention unless there is some possible explanation or at least statistically significant( p value usually used is <0.05 to say statistically significant).

Line 157- Table 1 -please indicate the lowest age group in the range.

268-270 The sentence “which may be attributed to the increased manipulation of samples resulting in an increased risk of contamination and labelling errors and these varying results may reflect the competency and dedicated time available of laboratory staff for sample processing.”

How did you reach to this argument? Was it stated by the studies as their limitation?
---

## [Editor Report · Decision Letter 2]

19 Jan 2022

Pooling sputum for Xpert MTB/RIF and Xpert Ultra testing during the Covid-19 pandemic in Lao People's Democratic Republic

PGPH-D-21-00728R2

Dear Dr. Iem,

We're pleased to inform you that your manuscript has been judged scientifically suitable for publication and will be formally accepted for publication once it meets all outstanding technical requirements.

Within one week, you'll receive an e-mail detailing the required amendments. When these have been addressed, you'll receive a formal acceptance letter and your manuscript will be scheduled for publication.

An invoice for payment will follow shortly after the formal acceptance. To ensure an efficient process, please log into Editorial Manager at https://www.editorialmanager.com/pgph/ click the 'Update My Information' link at the top of the page, and double check that your user information is up-to-date. If you have any billing related questions, please contact our Author Billing department directly at authorbilling@plos.org.

Kind regards,

Miguel Angel Garcia-Bereguiain, PhD

Academic Editor
